# Human Cytomegalovirus Is Associated with Lower HCC Recurrence in Liver Transplant Patients

Po-Jung Hsu [1], Hao-Chien Hung [2], Jin-Chiao Lee [2], Yu-Chao Wang [2], Chih-Hsien Cheng [2], Tsung-Han Wu [2], Ting-Jung Wu [2], Hong-Shiue Chou [2], Kun-Ming Chan [2], Wei-Chen Lee [2] and Chen-Fang Lee [2,3,*]

[1] Department of General Surgery, Chang-Gung Memorial Hospital, Linkou 333, Taiwan; mr1661@cgmh.org.tw

[2] Department of Liver and Transplantation Surgery, Chang-Gung Memorial Hospital, Linkou 333, Taiwan; mp0616@cgmh.org.tw (H.-C.H.); b9302012@cgmh.org.tw (J.-C.L.); b9002072@cgmh.org.tw (Y.-C.W.); chengcchj@cgmh.org.tw (C.-H.C.); domani@cgmh.org.tw (T.-H.W.); wutj5056@cgmh.org.tw (T.-J.W.); chouhs@cgmh.org.tw (H.-S.C.); chankunming@cgmh.org.tw (K.-M.C.); weichen@cgmh.org.tw (W.-C.L.)

[3] College of Medicine, Chang-Gung University, Taoyuan 333, Taiwan

* Correspondence: lee5310@cgmh.org.tw; Tel.: +886-3-3281200 (ext. 3366); Fax: +886-3-3285818

**Abstract:** Human cytomegalovirus (CMV) infection has been reported to compromise liver transplantation (LT) outcomes. Recent studies have shown that CMV has a beneficial oncolytic ability. The aim of this study was to investigate the impact of CMV on tumor recurrence in patients with hepatocellular carcinoma (HCC) who underwent liver transplantation (LT). This retrospective study enrolled 280 HCC patients with LT at our institute between January 2005 and January 2016. Their relevant demographic characteristics, pre- and post-LT conditions, and explant histology were collected. A CMV pp65 antigenemia assay was performed weekly following LT to identify CMV infection. A total of 121 patients (43.2%) were CMV antigenemia-positive and 159 patients (56.8%) were negative. A significantly superior five-year recurrence-free survival was observed among CMV antigenemia-positive patients compared with the CMV-negative group (89.2% vs. 79.9%, $p = 0.049$). There was no significant difference in overall survival between the positive and negative CMV antigenemia groups (70.2% vs. 75.3%, $p = 0.255$). The major cause of death was HCC recurrence in CMV antigenemia-negative patients (51.3%), whereas more CMV antigenemia-positive patients died due to other bacterial or fungal infections (58.3%). In the multivariate analysis, the independent risk factors for tumor recurrence included positive CMV antigenemia ($p = 0.042$; odds ratio (OR) = 0.44; 95% confidence interval (CI) = 0.20–0.97), microscopic vascular invasion ($p = 0.001$; OR = 3.86; 95% confidence interval (CI) = 1.78–8.36), and tumor status beyond the Milan criteria ($p = 0.001$; OR = 3.69; 95% CI = 1.77–7.71). In conclusion, in addition to the well-known Milan criteria, human CMV is associated with a lower HCC recurrence rate after LT. However, this tumor suppressive property does not lead to prolonged overall survival, especially in severely immunocompromised patients who are vulnerable to other infections.

**Keywords:** cytomegalovirus; hepatocellular carcinoma; liver transplantation

## 1. Introduction

Since the introduction of the Milan criteria (single tumor with diameter ≤ 5 cm, up to three tumors with diameter ≤ 3 cm, and no major vessel or extrahepatic involvement) [1], liver transplantation (LT) has served as one of the treatment choices for patients with an unresectable hepatocellular carcinoma (HCC). The five-year overall survival (OS) of patients with HCC who underwent LT exceeds 70% [1] Nevertheless, the reported HCC recurrence rate is 8–20% following transplantation [2–5]. Once HCC recurs, the estimated five-year overall survival decreases to 22–43% [2,4]. As a result, preventing HCC recurrence after LT remains an important issue, and many risk factors associated with tumor recurrence have been identified, such as tumor behavior, differentiation type, alpha-fetoprotein (AFP) level, and serum neutrophil-to-lymphocyte ratio [2,3].

Human cytomegalovirus (CMV), one of the most common opportunistic infections following transplantation, has been reported to increase the risk of allograft failure and compromise post-LT outcomes [6,7]. A high plasma CMV DNA load indicates a risk of developing major post-transplant complications [8]. Interestingly, recent studies have revealed that CMV has potential oncolytic activity, inducing apoptosis and stimulating immune cell infiltration in the tumor microenvironment [9]. Several human and animal models, based on various cancers, have demonstrated the anti-cancer ability of CMV in recent years [9–12]. Kumar et al. reported that CMV could limit tumor cell proliferation and enhance tumor cell apoptosis in a murine model [11].

Since clinical data showing this oncolytic effect are scarce, we are interested in investigating whether CMV can play a beneficial antitumor role after liver transplantation in the real world. The aim of this study was to clarify the impact of CMV on tumor recurrence and overall survival in HCC patients after LT. We have attempted to provide a novel viewpoint of viral infectious disease in this immunocompromised population.

## 2. Materials and Methods

### 2.1. Patient Enrollment and Data Collection

This was a retrospective study of consecutive LT cases with HCC at Chang Gung Memorial Hospital at Linkou from 2005 to 2016. We excluded patients who died early post-transplantation (within 90 days) or who had no sufficient follow-up period to monitor the primary outcome or tumor recurrence. Patients who were missed for follow-up or had incomplete data were also excluded. A total of 280 patients were enrolled in the study. Their relevant demographic characteristics, preoperative conditions, post-LT outcomes, explant histology, and CMV antigenemia data were collected. The protocol of this retrospective study was approved by the Ethics Committee and Institutional Review Board of Chang Gung Memorial Hospital (approval no. 202101491B0) and conformed to the ethical guidelines of the 1975 Declaration of Helsinki.

### 2.2. Liver Transplant Protocol and HCC Patients Selection

The pre-transplant evaluation, preparation, and procedures of liver transplantation were reported in our previously published studies [13–15]. Most of our patients received the right lobe from living donors they were related to. Milan criteria were used for the selection of HCC patients who were planning to receive LT. However, tumor status not beyond the University of California San Francisco (UCSF) criteria (solitary tumor ≤ 6.5 cm or up to three tumors ≤ 4.5 cm) [16] was also allowed. In our institute, the primary immunosuppression protocol consists of administering tacrolimus, corticosteroids, and mycophenolate mofetil. The dosage and titration protocols are described elsewhere [15].

### 2.3. CMV Serological Study and Standardization of CMV Surveillance

All recipients underwent serological tests for CMV antibodies (IgM and IgG) using an enzyme-linked immunosorbent assay before live transplantation. We routinely performed the CMV pp65 antigenemia assay weekly following liver transplantation until the patient was discharged or died. The present study did not include CMV quantitative polymerase chain reaction (qPCR) results because CMV qPCR has only been performed in our hospital since 2017.

### 2.4. Definition of CMV pp65 Antigenemia and CMV Disease

The protocol for the CMV pp65 antigenemia assay was documented in our previous study [8]. In brief, a blood sample was collected and an antigenemia assay was conducted within 6 h using the MonoFluoTM Kit CMV 52206 Immunofluorescence Assay (IFA; Bio-Rad, Hercules, CA, USA). CMV pp65 was targeted by a monoclonal antibody in the kit and visualized with a fluorescent secondary antibody. Positive CMV pp65 antigenemia was defined as at least one CMV pp65 antigen targeted by a monoclonal antibody per $500 \times 10^3$ peripheral blood leukocytes.

CMV disease was defined as the coexistence of documented positive CMV pp65 antigenemia and clinical symptoms, such as unexplained fever, thrombocytopenia ($<150 \times 10^3/\mu L$), leukopenia ($<4000/\mu L$), and/or atypical lymphocytosis ($>5\%$) [17]. In the present study, we defined severe CMV disease as two or more organ systems involved when CMV disease occurred and eventually led to organ failure [18].

### 2.5. Preemptive Treatment Protocol for CMV

Based on their clinical conditions, patients started anti-CMV treatment with oral valganciclovir 900 mg per day or intravenous ganciclovir 5 mg/kg twice per day once CMV antigenemia was detected, and treatment continued until the CMV pp65 antigenemia assay was negative.

### 2.6. Post-Transplant Outcome Assessment

Infection, acute cellular rejection, surgical complications, and tumor recurrence were the major post-LT events recorded in our study. The diagnosis of infection was based on positive culture results of blood, urine, ascites, or sputum specimens. We did not routinely perform liver allograft biopsy; acute rejection was defined as an elevation of > 30 IU/L of serum aspartate aminotransferase (AST) and alanine aminotransferase (ALT) within 24 h, not due to other causes of hepatic transaminase elevation [19]. A major post-transplant complication was defined as Clavien-Dindo class IV or V, which meant that patients experienced life-threatening organ dysfunction or even death [20]. HCC recurrence was diagnosed when the tumor recurred in the liver graft or any place in the body after a period when the cancer could not be detected. For tumor surveillance after transplantation, serum AFP and Doppler ultrasound were performed every three months along with computed tomography (CT) every six months, or when suspicious liver nodules were detected by ultrasound or with rising serum AFP. Recurrence-free survival (RFS) and overall survival were the secondary outcomes of the study and were calculated from the day of transplantation to the date of tumor recurrence or death.

### 2.7. Statistical Analysis

Demographic characteristics were summarized as median values, mean values ± standard deviations, or numbers with percentages. Categorical variables were compared between antigenemia-positive and antigenemia-negative patients using Pearson's chi-square test. Logistic regression analysis was used to predict the recurrence of HCC after liver transplantation. All potential variables identified in the univariate analysis ($p < 0.010$) were included in the multivariate model and utilized for backward selection. The five-year recurrence-free survival and five-year OS were compared using the Kaplan–Meier method. A two-tailed $p$-value of less than 0.05 was considered statistically significant. All statistical analyses were performed using SPSS statistics (version 22.0; SPSS Inc., Chicago, IL, USA).

## 3. Results

### 3.1. Characteristics of the Entire Population

The demographic data of 280 patients with HCC who underwent liver transplantation are summarized in Table 1. Most recipients were male ($n$ = 221, 78.9%), and the median age in the overall cohort was 56 (mean value: 56 ± 7.1) years. The median value of the MELD (model for end-stage liver disease) score was 12 (mean value: 13.5 ± 6.1). The main etiologies of liver disease are hepatitis B (HBV) and hepatitis C (HCV) infection. Among all patients, 173 (61.8%) had HBV and 79 (28.2%) had HCV. Most of the patients underwent living donor liver transplantation ($n$ = 233, 83.2%) and received right lobe grafts ($n$ = 220, 94.4%) with a median graft-to-recipient-weight ratio (GRWR) of 0.92% (mean value: 0.98 ± 0.22%). Before liver transplantation, 212 (75.7%) HCC patients received bridging or downstaging locoregional treatment, including transarterial chemoembolization in 195 patients (69.6%), radiofrequency ablation in 34 patients (12.1%), and other treatments (percutaneous ethanol injection and radiotherapy) in 20 patients (7.1%).

**Table 1.** Demographic characteristics of 280 HCC patients underwent LT.

| Factors | Median Value or Number (Percentage) | Mean ± SD | Range |
|---|---|---|---|
| General characteristics | | | |
| Recipient age | 56 | 56 ± 7.1 | 33–70 |
| Recipient gender, male | 221 (78.9%) | | |
| Pre-LT characteristics | | | |
| MELD score | 12 | 13.5 ± 6.1 | 5–40 |
| Hepatitis B infection | 173 (61.8%) | | |
| Hepatitis C infection | 79 (28.2%) | | |
| LDLT | 233 (83.2%) | | |
| Right lobe in LDLT | 220 (94.4%) | | |
| GRWR (%) in LDLT | 0.92 | 0.98 ± 0.22 | 0.57–1.79 |
| Local regional treatment before LT | 212 (75.7%) | | |
| Tumor status within Milan criteria (by radiologic assessment) | 234 (83.6%) | | |
| AFP | 13.4 | 213.5 ± 1168.2 | 1–18,250 |
| Explant pathology characteristics | | | |
| Recipient with solitary tumor | 109 (38.9%) | | |
| Maximum tumor size(cm) | 2.4 | 2.8 ± 1.6 | 0–11 |
| Satellite nodules | 24 (8.6%) | | |
| Macroscopic vascular invasion | 17 (6.1%) | | |
| Microscopic vascular invasion | 52 (18.6%) | | |
| CMV study | | | |
| Preoperative CMV IgG positive | 278 (99.3%) | | |
| PP65 antigenemia positive | 121 (43.2%) | | |
| PP65, maximum/per $500 \times 10^3$ PBL | 2 | | 1–115 |
| Persistent antigenemia > 2 weeks | 28/121 (23.1%) | | |
| Relapsed CMV antigenemia | 33/121 (27.3%) | | |
| Severe CMV disease | 6/121 (5%) | | |
| Clinical outcome | | | |
| Follow-up period(months) | 82.5 | 84.3 ± 49.4 | 3–191 |
| Five-year recurrence free survival, cumulative | 83.7% | | |
| Five-year overall survival, cumulative | 73.1% | | |
| Major complications | 23 (8.2%) | | |
| Cause of mortality in 5 years | | | |
| Other bacterial or fungal Infection | 30/75 (40.0%) | | |
| HCC-related | 26/75 (34.7%) | | |
| Rejection | 8/75 (10.7%) | | |
| Others | 11/75 (14.6%) | | |

Abbreviation: HCC, hepatocellular carcinoma; LT, liver transplantation; SD, standard deviation; MELD, model of end-stage liver disease; LDLT, living donor liver transplantation; GRWR, graft recipient weight ratio; AFP, alpha-fetoprotein; CMV, cytomegalovirus; PBL, peripheral blood leukocytes.

Preoperative liver CT was routinely performed, and 234 patients (83.6%) fulfilled the Milan criteria. The median value of pre-LT AFP was 13.4 ng/mL. After reviewing all corresponding explant histology, 109 patients (38.9%) had a solitary tumor, and the median maximum tumor size was 2.4 cm (mean value: 2.8 ± 1.6 cm). The presence of satellite nodules (*n* = 24, 8.6%), macroscopic vascular invasion (*n* = 17, 6.1%), and microscopic vascular invasion (*n* = 52, 18.6%) were documented. Regarding the preoperative CMV serologic test, 278 patients (99.8%) had positive CMV IgG results, and none of them had simultaneous positive CMV IgM. The CMV pp65 antigenemia assay was performed weekly after liver transplantation, and 121 patients (43.2%) experienced one or more positive results in serial examinations. The maximum number of visualized pp65 counts per 500,000 in the weekly test among CMV antigenemia-positive patients was recorded, with a median value of 2. Those cases with persistent antigenemia for more than two weeks and relapsed CMV antigenemia during the treatment course accounted for 23.1% (*n* = 28) and 27.3%

($n$ = 33), respectively. Despite the administration of anti-CMV regimens, six patients (5%) developed severe CMV disease.

The median follow-up time was 82.5 months (mean: 84.3 ± 49.4). A total of 75 patients (26.8%) expired within five years after LT, and the major causes of mortality were infection ($n$ = 30, 40%) and HCC recurrence ($n$ = 26, 34.7%). The five-year RFS and five-year OS rates in our cohort were 83.7% and 73.1%, respectively.

### 3.2. Comparison between CMV Antigenemia-Positive and Negative Patients

In this study, 121 patients had positive CMV antigenemia assays, and 159 patients were CMV antigenemia-negative. Comparisons of pre-LT demographic characteristics, HCC histological features, and clinical outcomes between these two groups are summarized in Table 2. Patients with and without CMV antigenemia were similar in terms of age, sex, surgical course, and explant pathologic findings. However, the CMV antigenemia-positive patients had higher MELD scores, indicating a weakened condition before transplantation in this population. Regarding HCC histological features, there were no differences in tumor number, tumor size, vascular invasion, and Milan criteria fulfillment between the two groups. Most strikingly, the five-year recurrence-free survival rate of HCC was significantly different between groups (Figure 1): 89.2% in CMV antigenemia-positive patients, as compared with 79.9% in CMV antigenemia-negative patients ($p$ = 0.049). However, there was no significant difference in the cumulative five-year OS between the two groups (70.2% vs. 75.3%, $p$ = 0.255, Figure 2). In addition, the antigenemia-positive group had more major surgical complications (14.9% vs. 3.2%, $p$ < 0.001). We further analyzed the causes of death and found that the CMV antigenemia-negative patients had a larger proportion of HCC recurrence-related mortality (16.7% vs. 51.3%, $p$ = 0.002), whereas the CMV antigenemia-positive patients faced more deaths from other bacterial or fungal infections (58.3% vs. 23.1%, $p$ = 0.002). These results highlight that CMV is indeed associated with lower tumor recurrence. However, severely compromised immunity increases the infection and complication rates that impede longer overall survival in the CMV-positive group.

### 3.3. Univariate and Multivariate Logistic Regression for Predictors of HCC Recurrence

Univariate and multivariate analyses were used to evaluate the potential predictors of HCC recurrence. The following clinical variables were used for univariate analysis: recipient age (>60 years), recipient sex, MELD score (>20), HBV infection, HCV infection, transplant type, pre-LT AFP (>200 ng/mL), pre-LT locoregional therapy, Milan criteria fulfillment, explanted liver characteristics, and CMV antigenemia. Significant results are shown in Table 3. Tumor status beyond the Milan criteria (either by radiological or histological assessment), positive CMV antigenemia assay, and histological factors such as multiple tumors, maximum tumor size > 3 cm, satellite nodules, and vascular invasion, were subsequently considered as potential risk factors for HCC recurrence within five years in the univariate logistic regression model. These potential risk factors were entered into the multivariate analysis, showing that positive CMV antigenemia ($p$ = 0.042; odds ratio (OR)= 0.44; 95% confidence interval (CI) = 0.20–0.97), microscopic vascular invasion ($p$ = 0.001; OR = 3.86; 95% CI= 1.78–8.36), and tumor status beyond the Milan criteria (explant) ($p$ = 0.001; OR = 3.69; 95% CI = 1.77–7.71) were independent risk factors for HCC recurrence. Based on our results, the Milan criteria remains the gold standard for patient selection, and we put forward a new insight that human CMV is associated with lower HCC recurrence following LT.

**Table 2.** Demographic characteristic according to PP65 antigenemia positive and negative.

| Factors | CMV Positive *n* = 121 | CMV Negative *n* = 159 | *p*-Value |
|---|---|---|---|
| General characteristic | | | |
| Recipient age, year-old (>60) | 40 (33.1%) | 46 (28.9%) | 0.458 |
| Recipient gender, male | 94 (77.7%) | 127 (79.9%) | 0.656 |
| Pre-LT characteristic | | | |
| MELD score >20 | 23 (19.0%) | 11 (6.9%) | 0.002 |
| Hepatitis B infection | 70 (57.9%) | 103 (64.8%) | 0.237 |
| Hepatitis C infection | 39 (32.2%) | 40 (25.2%) | 0.193 |
| LDLT | 96 (79.3%) | 137 (86.2%) | 0.130 |
| Right lobe in LDLT | 89/96 (92.7%) | 131/137 (95.6%) | 0.340 |
| GRWR ≤ 0.8% in LDLT | 20/96 (20.8%) | 27/137 (19.7%) | 0.833 |
| Local regional treatment before LT | 89 (73.6%) | 123 (77.4%) | 0.462 |
| Beyond Milan criteria | 21 (17.4%) | 25 (15.7%) | 0.715 |
| AFP > 200 ng/mL | 18 (14.9%) | 22 (13.8%) | 0.805 |
| Explant pathology characteristic | | | |
| Recipients with multiple tumors | 77 (63.6%) | 91 (59.1%) | 0.443 |
| Maximum tumor size > 3 cm | 39 (32.2%) | 52 (32.7%) | 0.933 |
| Satellite nodules | 7 (5.8%) | 17 (10.7%) | 0.146 |
| Macroscopic vascular invasion | 6 (5.0%) | 11 (6.9%) | 0.496 |
| Microscopic vascular invasion | 23 (19%) | 29 (18.2%) | 0.870 |
| Beyond Milan criteria | 37 (30.6%) | 60 (37.7%) | 0.212 |
| Clinical outcome | | | |
| Five-year recurrence free survival, cumulative | 89.2% | 79.9% | 0.049 |
| Five-year overall survival, cumulative | 70.2% | 75.3% | 0.255 |
| Major complications | 18 (14.9%) | 5 (3.2%) | <0.001 |
| Causes of death in 5 years after LT | | | 0.004 |
| Other bacterial or fungal Infection | 21 (58.3%) | 9 (23.1%) | 0.002 |
| HCC related | 6 (16.7%) | 20 (51.3%) | 0.002 |
| Rejection | 5 (13.9%) | 3 (7.7%) | 0.385 |
| Others | 4 (11.1%) | 7 (17.9%) | 0.403 |

Abbreviation: HCC, hepatocellular carcinoma; LT, liver transplantation; SD, standard deviation; MELD, model of end-stage liver disease; LDLT, living donor liver transplantation; GRWR, graft recipient weight ratio; AFP, alpha-fetoprotein; CMV, cytomegalovirus; PBL, peripheral blood leukocytes.

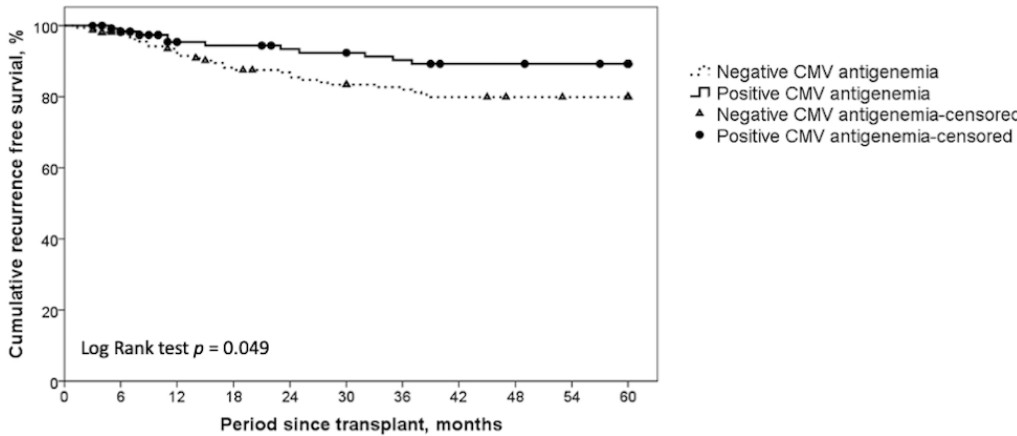

**Figure 1.** Kaplan–Meier method for five-year cumulative recurrence-free survival (RFS) depending on CMV antigenemia positivity and negativity. Positive CMV antigenemia group showed a significantly superior five-year RFS.

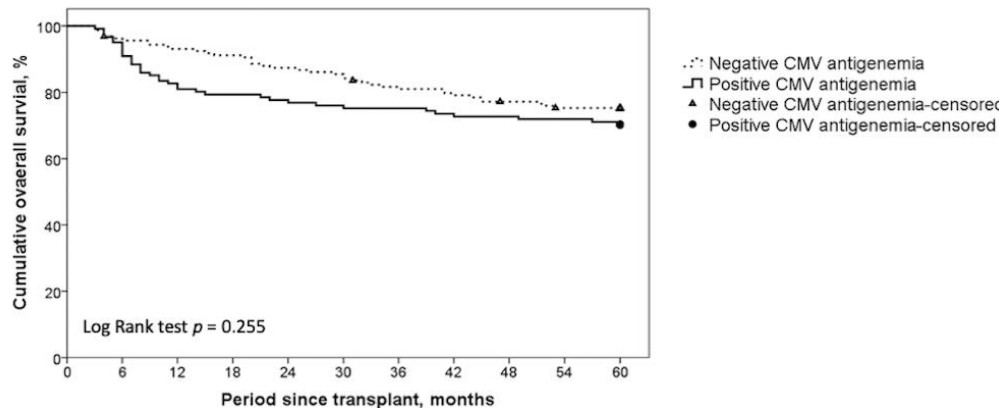

**Figure 2.** Kaplan–Meier method for five-year cumulative overall survival (OS) depending on positive and negative CMV antigenemia results. The five-year cumulative OS between the CMV antigenemia-positive and CMV antigenemia-negative groups were not significantly different.

**Table 3.** Uni-/multivariate analyses in predicting HCC recurrence after liver transplantation.

| Factors | Univariate | | | Multivariate | | |
|---|---|---|---|---|---|---|
| | OR | 95%CI | *p*-Value | OR | 95%CI | *p*-Value |
| Pre-LT characteristic | | | | | | |
| Beyond Milan criteria (by radiology) | 3.35 | 1.59–7.07 | 0.001 | | | |
| CMV study | | | | | | |
| Positive CMV antigenemia | 0.43 | 0.21–0.90 | 0.025 | 0.44 | 0.20–0.97 | 0.042 |
| Explant pathology characteristic | | | | | | |
| Multiple tumor numbers | 2.56 | 1.17–5.60 | 0.019 | | | |
| Maximum tumor size > 3 cm | 2.85 | 1.45–5.60 | 0.002 | | | |
| Satellite nodule | 3.38 | 1.34–8.51 | 0.010 | | | |
| Macroscopic vascular invasion | 3.55 | 1.24–10.22 | 0.019 | | | |
| Microscopic vascular invasion | 4.72 | 2.31–9.65 | <0.001 | 3.86 | 1.78–8.36 | 0.001 |
| Beyond Milan criteria (by pathology) | 4.10 | 2.05–8.21 | <0.001 | 3.69 | 1.77–7.71 | 0.001 |

Abbreviation: OR, odds ratio; CI, confidence interval; progressive disease; LT, liver transplantation; HCC, hepatocellular carcinoma; AFP, alpha-fetoprotein; CMV, cytomegalovirus. All the factors calculated in UV were put in MV analysis with backward stepwise (Likelihood ratio).

## 4. Discussion

Liver transplantation is known to be a curative treatment option for HCC, especially for patients who are cirrhotic or not eligible for hepatectomy. Post-LT five-year OS was reported to reach 76% in recipients meeting the Milan criteria [1]. Some groups have proposed expansion beyond the Milan criteria. For example, the UCSF and up to seven criteria [21,22] have shown promising results. Therefore, the number of patients receiving transplantation to treat HCC are increasing and avoiding tumor recurrence has become an important concern. Similar to others, our study found that the Milan criteria remain a good selection tool and histological findings are the main predictors. Importantly, we found that HCC patients with CMV antigenemia were associated with significantly higher five-year RFS after transplantation than those who test negative for CMV. Although this phenomenon did not provide a benefit for overall survival, the leading cause of mortality among CMV-negative patients was HCC recurrence. Meanwhile, the major cause of death in CMV-positive patients was other bacterial or fungal infections. Our study not only emphasizes the antitumor property of CMV in LT but also reminds us of the risk of the severely immunocompromised situation in CMV-positive patients receiving LT.

With its potential oncolytic ability, the pathophysiological modulation in the host microenvironment by CMV has been widely studied as a therapeutic possibility in various cancer treatments [23,24]. For example, CMV infection slows tumor proliferation and promotes intrinsic caspase cancer cell apoptosis in a murine HCC study [11]. Many

models have been designed to investigate the possible underlying mechanisms, and CMV is thought to induce tumor cell death by killing cells directly or stimulating cytokine and antitumor immune responses [9]. Studies have also reported that CMV could be used as a potential vaccine vector against cancers by eliciting a long-acting T-cell response [10,12,25], regardless of pre-existing anti-CMV cellular immunity [26]. Indeed, CMV offers some advantages that make it an attractive platform for anti-cancer studies. Regarding the immunosuppression protocol for CMV infection, we did not routinely adjust the immunosuppression for CMV antigenemia-positive patients because most of them were asymptomatic and were under preemptive treatment for CMV. We did lower the dose of steroid in patients with CMV disease. The trough levels of calcineurin inhibitor were kept at the same level (5 to 8 ng/mL) in two groups. Therefore, the difference of recurrence free survival was not related to the use of calcineurin inhibitor, which is known to be associated with de novo malignancy or recurrence of malignancy after solid organ transplantation. It is also worthy to mention that everolimus, an inhibitor of mammalian target of rapamycin, was reported to alleviate CMV infection [27]. Everolimus has anti-cancer and immunosuppressive effects and can be used for the prevention or treatment of HCC recurrence following LT [28,29]. The antiviral efficacy of everolimus may counteract the oncolytic effect of CMV in transplantation.

We do not advocate ignoring the negative impact of CMV on LT. CMV infection has been considered a major concern for post-transplant recipients. CMV disease is common in 8–29% of all LT recipients [30–32]. Progression of CMV disease often results in major organ failure and death [33,34]. This common opportunistic pathogen incurs a more complex infectious condition in immunocompromised recipients following organ transplantation [35,36]. We encourage aggressive CMV treatment once it is detected due to its lethality and many potential risks in transplant recipients [8]. However, if the clinical presentation of CMV infection becomes latent, we may have some room to make our treatment policy more flexible.

Our study has some limitations. First, compared to the CMV qPCR method, CMV pp65 antigenemia is not an ideal quantitative method for detecting CMV infection [37–39]. Second, we performed the CMV pp65 antigenemia assay weekly during hospitalization, and a longer period of CMV surveillance may be needed. Third, this study was conducted at a single center and was consequently subjected to its bias of CMV treatment and monitoring protocol. Accordingly, further prospective, and multi-center research with the CMV qPCR method may be required to verify our results.

## 5. Conclusions

In conclusion, CMV reduces the risk of HCC recurrence following LT. This tumor suppressive property does not lead to prolonged overall survival, especially in severely immunocompromised patients who are vulnerable to other infections. Nevertheless, the potential anti-HCC benefit we observed in survivors of CMV infection after LT deserves further investigation. Our study provides a new viewpoint, offering a magnificent possibility for oncolytic viruses in transplantation.

**Author Contributions:** P.-J.H. participated in the writing of the paper, research design, performance of the research, and data analysis. H.-C.H., J.-C.L., Y.-C.W., C.-H.C., T.-H.W., T.-J.W., H.-S.C. and K.-M.C. participated in research design, and performance of the research. W.-C.L. and C.-F.L. participated in research design and data analysis. All authors have read and agreed to the published version of the manuscript.

**Funding:** This research received no external funding.

**Institutional Review Board Statement:** The study was conducted according to the guidelines of the Declaration of Helsinki and approved by the Institutional Review Board of Chang-Gung Memorial Hospital (IRB No. 202101491B0).

**Informed Consent Statement:** Patients were not required to give informed consent to the study because the analysis used anonymous clinical data that were obtained after each patient agreed to treatment by written consent.

**Data Availability Statement:** The data presented in this study are available on request from the corresponding author.

**Acknowledgments:** The authors gratefully acknowledge all LT team members who dedicate their best efforts in taking care of transplant patients in our hospital.

**Conflicts of Interest:** The authors declare no conflict of interest.

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
