# Peer review of "Human Cytomegalovirus Is Associated with Lower HCC Recurrence in Liver Transplant Patients"

_curroncol, doi:10.3390/curroncol28060364_

Round 1

Reviewer 1 Report

Very interesting and well written manuscript. I have two small comments:

  • Paragraph 2.3. There is part of a sentence at the end that does not belong there
  • Line 233. Take a look at the sentence; you mean the opposite.

Reviewer 2 Report

The topic is highly interested for clinicians. Most of surgeons and oncologists used to find CMV infection as deadly cofactor in tumor and transplants outcome, however Authors proved it is not the case in patients who underwent liver transplantation due to HCC. Moreover they tried to explain potential benefits from CMV infection, particularly on the cell biology level.

The manuscript is well organized and contains up-to-date introduction, clear description of material and methods, interesting results and excellent discussion. The limitations of this study are well defined.

There is a number of style and editorial mistakes.

Line 86 - unwanted last sentence

Line 94 - incorrect number 103

Table 1. AFP value?, median?, what about SD?

Table 1. Tumor number, solitary - is this a percentage of patients?

The main disadvantage of this study is use of PP65 instead of CMV-DNA, but it has been clearly explained due to period (years) of observation.

I would recommend to accept the manuscript after these minor mistake correction

Reviewer 3 Report

The aim of this study was to investigate the impact of CMV on tumor recurrence in patients with 13 hepatocellular carcinoma (HCC) who underwent liver transplantation.

This study is well -organized. However, I have some comments on this study.

  1. As shown in Table 1, there were 99.3% (278/280) of recipients’ serum CMV IgG were positive. Thus, what is the explanation that the positive rate of serum PP65 antigenemia of all patients of this study is 43.2% (121/280)? The rate of 43.2% is significantly higher than of that reported in Taiwan (Transplant Proc. 2014;46(3):832-4)

2.Is there any difference existing in protocol of adjustment of immunosuppression in patient with or without positive serum PP65 antigenemia? As shown in Table 2, there were 21 (58.3%) and CMV antigenemia-positive patients died due to infection and only 9 (23.1%) CMV antigenemia-negative patients died due to infection(p=0.002). Therefore, did authors decrease the dosage of immunosuppressants, especially calcineurin inhibitor, in CMV antigenemia-positive patients? If so, did lower dose or lower serum level of calcineurin inhibitor play a more important role than that of CMV antigenemia in decrease of incidence of recurrence of HCC in CMV antigenemia-positive patients?

Because calcineurin inhibitor plays an important role in development of de novo malignancy or recurrence of malignancy after solid organ transplantation, authors may provide the data of trough level of tacrolimus of CMV antigenemia-positive/ negative patients enrolled in this study.

  1. Did the “infection” shown in Table 1 specific to CMV infection or mean other systemic infection such as pneumonia, urinary tract infection…? Therefore, authors may provide the data of CMV infection, CMV disease, and asymptomatic CMV infection in CMV antigenemia-positive patients and their impact on recurrence of HCC.   
  2. As shown in Table 2, the 5-year recurrence free survival of both CMV antigenemia-positive/ negative patients are higher than their 5-year overall survival in this study.

  It makes readers puzzled.      
